# Development of a High-Resolution Single-Nucleotide Polymorphism Strain-Typing Assay Using Whole Genome-Based Analyses for the *Lactobacillus acidophilus* Probiotic Strain

**DOI:** 10.3390/microorganisms8091445

**Published:** 2020-09-21

**Authors:** Chien-Hsun Huang, Chih-Chieh Chen, Shih-Hau Chiu, Jong-Shian Liou, Yu-Chun Lin, Jin-Seng Lin, Lina Huang, Koichi Watanabe

**Affiliations:** 1Bioresource Collection and Research Center, Food Industry Research and Development Institute, 331 Shih-Pin Rd, Hsinchu 30062, Taiwan; chh@firdi.org.tw (C.-H.H.); shc@firdi.org.tw (S.-H.C.); ljs@firdi.org.tw (J.-S.L.); hln@firdi.org.tw (L.H.); 2Institute of Medical Science and Technology, National Sun Yat-sen University, Kaohsiung 80424, Taiwan; chieh@imst.nsysu.edu.tw; 3Rapid Screening Research Center for Toxicology and Biomedicine, National Sun Yat-sen University, Kaohsiung 80424, Taiwan; 4Livestock Research Institute, Council of Agriculture, Executive Yuan, Tainan 71246, Taiwan; hiujj@tlri.gov.tw; 5Culture Collection & Research Institute, Synbio Tech Inc., Kaohsiung 82151, Taiwan; jslin@synbiotech.com.tw; 6Department of Animal Science and Technology, College of Bioresources and Agriculture, National Taiwan University, No. 50, Ln. 155, Sec. 3, Keelung Rd., Taipei 10673, Taiwan

**Keywords:** whole genome sequences, core genome MLST, Strain-Specific Identification, monophyletic bacterium, *Lactobacillus acidophilus*

## Abstract

*Lactobacillus acidophilus* is one of the most commonly used industrial products worldwide. Since its probiotic efficacy is strain-specific, the identification of probiotics at both the species and strain levels is necessary. However, neither phenotypic nor conventional genotypic methods have enabled the effective differentiation of *L. acidophilus* strains. In this study, a whole-genome sequence-based analysis was carried out to establish high-resolution strain typing of 41 *L. acidophilus* strains (including commercial isolates and reference strains) using the cano-wgMLST_BacCompare analytics platform; consequently, a strain-specific discrimination method for the probiotic strain LA1063 was developed. Using a core-genome multilocus sequence-typing (cgMLST) scheme based on 1390 highly conserved genes, 41 strains could be assigned to 34 sequence types. Subsequently, we screened a set of 92 loci with a discriminatory power equal to that of the 1390 loci cgMLST scheme. A strain-specific polymerase chain reaction combined with a multiplex minisequencing method was developed based on four (*phoU*, *secY*, *tilS*, and *uvrA_*1) out of 21 loci, which could be discriminated between LA1063 and other *L. acidophilus* strains using the cgMLST data. We confirmed that the strain-specific single-nucleotide polymorphisms method could be used to quickly and accurately identify the *L. acidophilus* probiotic strain LA1063 in commercial products.

## 1. Introduction

*Lactobacillus acidophilus* is a commonly recognized species of lactic acid bacteria (LAB) that can be isolated from animal and human microbiota, such as those of the feces, mouth, and vagina [1,2,3,4]. *L. acidophilus* strains have been widely used in commercial probiotic products, including cheese, acidophilus milk, and yogurt, as well as in dietary supplements, with reported functional effects [5]. *L. acidophilus* NCFM is a well-known probiotic strain that is generally recognized as safe by the United States Food and Drug Administration: it improves the human intestinal environment and adjusts the balance of enteric bacteria [6]. The health benefits attributed to probiotic microorganisms are strain-specific [7,8]. Huys et al. [9] indicated that, because of methods that limit taxonomic resolution, more than 28% of commercially available probiotics are incorrectly labeled at the species or genus level. The ability to accurately identify probiotic strains in products is critical for suppliers and manufacturers. Therefore, starter cultures must be identified not only at the species level, but also, at the strain level to manage and control the quality of probiotic products.

Conventional molecular methods for strain typing *L. acidophilus*, such as randomly amplified polymorphic DNA (RAPD), matrix-assisted laser desorption/ionization–time-of-flight mass spectrometry, and pulsed-field gel electrophoresis (PFGE), are based on DNA and protein fingerprinting [10,11,12]. Although PFGE is considered the gold standard for bacterial strain typing, the discriminatory power of these methods is insufficient [13]. Multilocus sequence typing (MLST) is based on partial nucleotide sequences of multiple housekeeping genes and has been used for *Lactobacillus* species, including *L. delbrueckii*, *L. fermentum*, *L. plantarum*, *L. paracasei*, and *L. sakei* [14,15,16,17,18]. MLST is a suitable alternative to PFGE [19]. Ramachandran et al. [11] reported that the MLST scheme using seven conserved housekeeping genes (*fusA*, *gpmA*, *gyrA*, *gyrB*, *lepA*, *pyrG*, and *recA*) could be used as an intraspecies subtyping technique for the *Lactobacillus* complex (*L. acidophilus*, *L. amylovorus*, *L. crispatus*, *L. gallinarum*, *L. gasseri*, and *L. johnsonii*); however, two allelic profiles from five *L. acidophilus* strains were observed only in the *gyrA* gene. *L. acidophilus* strains have been considered to have little genome sequence variation [20,21,22,23], and they are regarded as a monophyletic taxon [10]. Therefore, higher-resolution strain-level differentiation methods must be developed for *L. acidophilus* strains.

With the technological achievement of whole-genome sequencing (WGS), dry-lab in silico analyses now rely on comparative genome sequences instead of conventional taxonomic methods for deep-level phylogenies [24,25]. Chun et al. [26] proposed minimum standards for species identification on the basis of overall genome-related indices, such as digital DNA‒DNA hybridization (dDDH) values and average nucleotide identity (ANI). By contrast, WGS-based strain typing uses a gene-by-gene approach, such as whole-genome MLST (wgMLST) or core-genome MLST (cgMLST), which entails the use of numerous gene loci to compare genomes [27,28]. These approaches provide resolutions superior to those of current subtyping techniques (including PFGE and multilocus variable-number tandem-repeat analysis (MLVA)) because they can discriminate between closely related strains of clinically relevant foodborne pathogens [29,30,31].

In this study, we aimed to develop a high-resolution strain-typing method for *L. acidophilus* probiotic strains, including a differential cgMLST scheme and a strain-specific detection technique, using comparative genome analyses.

## 2. Materials and Methods

### 2.1. L. acidophilus Strains and Culture Conditions

The 11 reference strains and probiotic strain LA1063 used in this study were obtained from the Bioresource Collection and Research Center (BCRC, Hsinchu, Taiwan), and Synbio Tech Inc., Kaohsiung, Taiwan, respectively, and they were authenticated through 16S rRNA gene sequencing (Appendix A). Strain LA1063 was isolated from feces of healthy Taiwanese adults and was used as a manufacturing strain for the probiotic supplements. The *Lactobacillus* strains were incubated anaerobically on Lactobacilli MRS agar (Difco Laboratories, Detroit, MI, USA) at 37 °C for 36 h, and fresh cultures were used for further DNA analyses.

### 2.2. WGS and Phylogenomic Metric Calculation

Genomic DNA was extracted using the EasyPrep HY genomic DNA extraction kit (Biotools Co. Ltd., Taipei, Taiwan) following the manufacturer’s protocols. The draft genomes of nine reference strains (BCRC 12255, BCRC 14065, BCRC 14079, BCRC 16092, BCRC 16099, BCRC 17008, BCRC 17481, BCRC 17486, and BCRC 80064) and the probiotic strain LA1063 were sequenced from an Illumina paired-end library with an average insert size of 350 bp by using an Illumina HiSeq4000 platform with the PE 150 strategy at Beijing Novogene Bioinformatics Technology Co., Ltd. (Beijing, China). The resulting raw reads were assembled de novo using SOAPdenovo software [32]. A total of 31 public genome sequences of *L. acidophilus* strains were downloaded from the United States National Center for Biotechnology Information bacterial genome database. Whole-genome similarities among the *L. acidophilus* strains were estimated using orthologous ANI [33].

### 2.3. cgMLST Scheme for L. acidophilus Strains

The cano-wgMLST_BacCompare web-based tool [34] was applied to the cgMLST analysis. This platform is composed of two major processes: whole-genome scheme extraction and discriminatory loci refinement. In this pipeline, genes were annotated using Prokka [35], and comparative genomics was performed using Roary [36]. Allele calling was performed using the Basic Local Alignment Search Tool Type N [37], with a minimum identity of 90% and coverage greater than 90% for the locus assignment (presence/absence profile) and exact match for the allele assignment (allele profile). Genetic relatedness trees were constructed from the allelic profile by using a neighbor-joining clustering algorithm in the Phylogeny Inference Package program [38], FigTree software (v1.4.3) [39], and GrapeTree software (v1.5.0) [40]. Finally, the Environment for the Tree Exploration v3 toolkit [41] and feature importance [42] program from scikit-learn [43] were applied to determine the discriminatory loci.

### 2.4. Validation of the cgMLST through a Differential MLST Scheme in Reference Strains

The cgMLST analysis results were validated by using an MLST scheme based on the reference strains. In short, by comparing with the sequence of cgMLST loci, the degenerate primers of several differential target genes were designed and tested (Appendix A). The 11 reference strains were used for validation. Polymerase chain reactions (PCRs) were performed using 81 μL of sterile Milli-Q water, 10 μL of 10× PCR buffer, 1.5 μL of denucleoside triphosphates (10 mM), 2.5 μL of forward primer (10 mM), 2.5 μL of reverse primer (10 mM), 2.5 U of *Taq* DNA polymerase (DreamTaq, Thermo Scientific, Waltham, MA, USA), and 3 μL of template DNA (100 ng/μL). The thermal protocol consisted of the following conditions: initial strand denaturation at 94 °C for 5 min, followed by 30 cycles at 94 °C for 1 min, 60 °C for 1 min, and 72 °C for 1 min, with a final extension step at 72 °C for 7 min. The resulting amplicons were purified using a QIA quick PCR Purification Kit (Qiagen Inc., Valencia, CA, USA) and sequenced using a BigDye Terminator v3.1 cycle-sequencing kit on a 3730 DNA Analyzer (Applied Biosystems and Hitachi, Foster City, CA, USA). The gene sequences of all strains obtained from sequencing were aligned using the Clustal X program, version 1.8 [44]. The MLST allele profiles and sequence types were analyzed using DnaSP version 5.1 [45].

### 2.5. Strain-Specific Identification for Probiotic Strain LA1063

The PCR- and single-nucleotide polymorphism (SNP)-based discrimination analyses were integrated for the direct strain-specific identification of probiotic strain LA1063. Strain-specific primers were designed using the genes that were chosen based on the presence or absence analysis and the cgMLST allele profiles. The multiplex minisequencing protocol for the analysis of SNPs was performed by following the method described by Huang et al. [46] and Lomonaco et al. [47]. The various concentrations of multiplex PCR and multiplex SNP-specific primers are listed in Table 1 and Table 2, respectively. The thermal cycling conditions for the multiple PCR and multiplex minisequencing were as follows: one cycle of 94 °C for 5 min; 30 cycles of 94 °C for 1 min, 60 °C for 1 min, and 72 °C for 1 min; one cycle of 72 °C for 7 min; and 25 cycles at 96 °C for 10 s, 50 °C for 5 s, and 60 °C for 1 min.

### 2.6. Authentication of Probiotic Strains in Commercial Products by Strain-Specific Assay

Three powder samples from separate batches of LA1063-derived materials for the production of probiotic supplements were analyzed. The LA1063 strain was isolated using serial dilution and plating methods and was identified using a MALDI Microflex LT mass spectrometer (Bruker Daltonics, Bremen, Germany), as described previously [48], followed by an LA1063 strain-specific assay.

## 3. Results and Discussion

The genomic data of all *L. acidophilus* strains had sequences of good quality that were directly reflected in the relatively small number of contigs (median, 24 and interquartile range, 17–34) (Appendix A), and these data were used in further comparative genomic approaches. Strains had diverse biochemical and phenotypic characteristics. However, high genome similarities among *L. acidophilus* strains are reported when the sequenced genomes are aligned [10]. This finding was consistent with the high ANI values (≥ 99.3%) among the 41 *L. acidophilus* strains in our study. In particular, most commercial isolates shared an extremely high degree of genome similarity (approximately 99.9%, Appendix A). Comparable genomic conservation levels were previously identified in *Bifidobacterium animalis* subsp. *lactis*, for which isolates from dissimilar commercial products had highly conserved genome sequences [49,50].

Comparative pan-genome analyses of LAB strains indicated that the health effects of those strains varied among species and strains [51,52,53,54,55]. This variation warrants the genome-level characterization of probiotic strains. A further analysis of genome sequences for high-resolution strain typing of 41 *L. acidophilus* strains was conducted using the cano-wgMLST_BacCompare analytics platform. For this dataset, the *L. acidophilus* pan-genome allele database (PGAdb) contained 2603 genes, of which 1390 (53.4%), 687 (26.4%), and 526 (20.2%) were core, accessory, and unique, respectively. Using the cgMLST analysis based on the allele profiles of the 1390 core genes, 38 out of 41 *L. acidophilus* strains were separated into two clusters, Cluster A (comprising 34 strains) and Cluster B (comprising four strains); moreover, Cluster A was further separated into three subclusters, namely Clusters A-1 (27 strains), A-2 (two strains), and A-3 (two strains), along with three disparate strains. Almost all strains (> 90%) in Cluster A-1 were the commercial strains (Figure 1). A total of 34 different sequence types (STs) were obtained from the 41 strains by using a minimum spanning tree based on the 1390 loci (Appendix A). Of these, 29 STs were assigned to single strains; four STs (ST5, ST19, ST27, and ST29) were assigned to two strains; and one ST (ST9) was assigned to four strains. A total of 22 out of the 26 STs comprising the commercial strains were grouped into a tight cluster. The differences between the strains within this tight cluster ranged from 0 to 53 alleles (Figure 2). Strain LA1063 showed 21 loci differences from strain BCRC 17481. This result demonstrates the extremely low diversity in commercial isolates and is consistent with the findings of a wgMLST study that used 1815 loci [10]. In addition, we screened a set of the 92 loci with a discriminatory power equal to that of the 1390 loci cgMLST scheme (Figure 3). Detailed information on these loci with high discriminatory powers is provided in Table 3. A differential MLST scheme based on the eight loci of *oppA*_1, *tr*, *ybhL*, *frdA*, hp, *rr*, *uvrA*_2, and *phoU* genes for 11 reference strains, and one probiotic strain (LA1063) was also used to validate the genome-based analytical data through direct Sanger sequencing. The partial sequences containing the informatic SNPs of the eight differential genes were successfully amplified and sequenced (data not shown) and could be distinguished into different STs, although their lengths were different from those of the WGS in the database (Table 4).

Compared with genome sequence–based typing, a rapid, precise, cost-efficient, and reproducible method for strain identification would be ideal for probiotic starter strains [56,57]. Strain-specific sequences for probiotic strains have been developed mainly by targeting the 16S–23S internal transcribed spacer region, phages, and protein-encoding genes [58,59,60], as well as by using DNA banding patterns [61,62,63,64,65,66,67,68]. However, because these methods have resolution limitations pertaining to monophyletic taxa, strain-specific marker identification can be replaced by a comparative genome analysis that targets unique insertions and deletions (INDELs) or SNPs in DNA sequences [47,69]. Gene presence/absence profiles among *L. acidophilus* strains were analyzed in a pan-genomic analysis by using 41 genomes, revealing that two strains (LA1063 and BCRC 17481) exhibited absences of the redox-sensing transcriptional repressor Rex 2 (*Rex*2) gene, whereas the other 39 strains had this gene (Figure 4). However, when we analyzed the specific primer pair targeted by *Rex2*, we found that strains LA1063 and BCRC 17481 had a 68-bp deletion in this gene (Appendix A).

To date, no consensus has been reached on the definition of a strain based on the number of nucleotide differences. However, a single-base pair cutoff has been discussed and considered by expert panels [57]. The 21 loci from the cgMLST data of 1390 core genes could be used to discriminate LA1063 from the other 40 strains (Table 5); therefore, to distinguish between LA1063 and other *L. acidophilus* strains, four of the 21 discriminated loci from the cgMLST data were selected, and they were confirmed to have T→G, T→G, T→G, and A→C nucleotide variations at the *phoU*, *secY*, *tilS*, and *uvrA_*1 loci, respectively (Appendix A).

Sharma et al. [70] successfully used RAPD, a repetitive element–based PCR method, and MLST for tracking intentionally inoculated LAB strains in yogurt and probiotic powder; their proposed polyphasic approach effectively tracked the starter strains. However, such an approach is time- and cost-intensive. By contrast, multiplex minisequencing can be used to identify the nucleotide located at a given site. This method is especially useful for simultaneously screening many SNPs within one reaction tube. Automated fluorescent capillary electrophoresis for minisequencing products can be performed in only 40 min, which is less than the 2.5 h required for direct sequencing. Multiplex minisequencing has been successfully developed for the identification and differentiation of probiotic bacteria at the strain, subspecies, and species levels [46,71,72,73].

Subsequently, multiplex minisequencing was performed to directly identify strain-specific SNP-based markers in the probiotic LA1063 strain. Primers specific to SNPs were used to achieve simultaneous annealing next to the nucleotide at strain-specific SNPs, and three of the primers included different-length 5′ nonhomologous poly(dGACT) tails to facilitate terminator-incorporated primer differentiation by size (Table 2). Subsequently, multiple PCR amplicons with four diagnostic sizes (165, 245, 255, and 457 bp) (Figure 5a) were purified and applied in multiplex minisequencing. Four peaks with expected colors and positions were observed for LA1063 (Figure 5b). Next, by using the strain-specific multiplex PCR and SNP primers, separate batches of LA1063-derived probiotics in powder products were analyzed, and the nucleotide bases were found to be identical to those of the original probiotic strain (Figure 5), which demonstrated the specificity and reproducibility of this method.

## 4. Conclusions

Conventionally, the strain-level typing and identification of monophyletic species such as *L. acidophilus* is challenging. Our study revealed that comparative genomic analysis could substantially increase discriminatory power to reach high-resolution typing on the basis of the allele profiles of the cgMLST scheme, and the 41 *L. acidophilus* strains were categorized into 34 STs. Consequently, the strain-specific identification method relying on INDEL-SNP markers was successfully developed and made available for the direct discrimination and tracking of probiotic strains in commercial products.

## Figures and Tables

**Figure 1 microorganisms-08-01445-f001:**
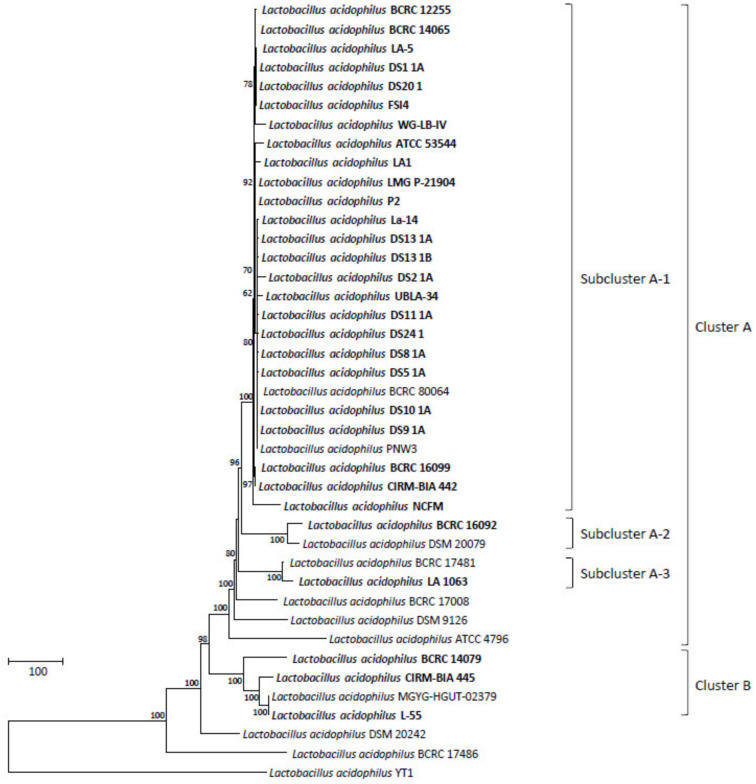
The allele-based neighbor-joining tree constructed with core-genome multilocus sequence-typing (cgMLST) profiles for the 41 *Lactobacillus acidophilus* strains on the basis of a comparison of 1390 differentiated core genes. The bold letters indicate commercial isolates. Bar, allele numbers.

**Figure 2 microorganisms-08-01445-f002:**
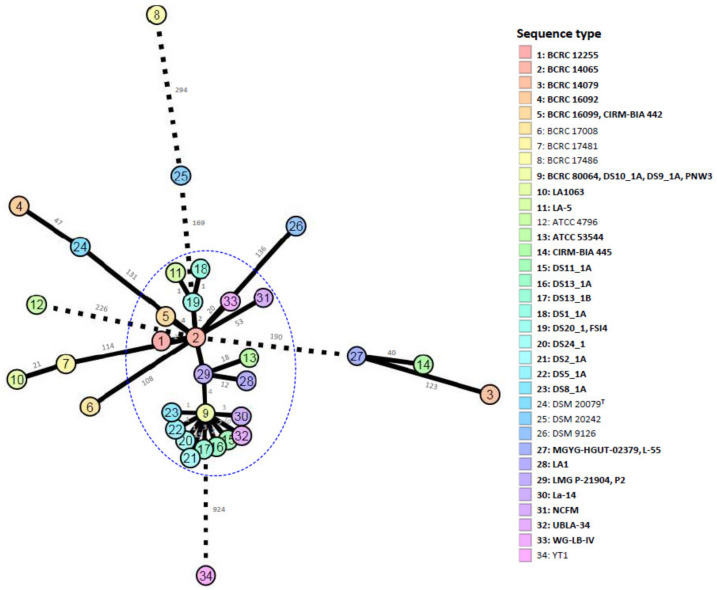
The allele-based minimum spanning tree constructed with cgMLST profiles for the 41 *Lactobacillus acidophilus* strains on the basis of a comparison of 1390 differentiated core genes. Each circle represents a different sequence type. Branch values indicate the number of loci that differ between nodes.

**Figure 3 microorganisms-08-01445-f003:**
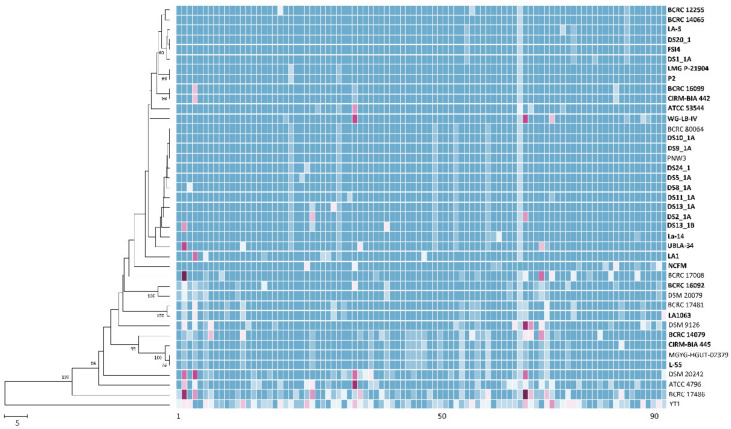
The allele-based neighbor-joining tree and heatmap constructed with cgMLST profiles for the 41 *Lactobacillus acidophilus* strains on the basis of 92 differentiated core genes. The bold letters indicate commercial strains. Different alleles in the same column are indicated by different colors. Bar, allele numbers.

**Figure 4 microorganisms-08-01445-f004:**
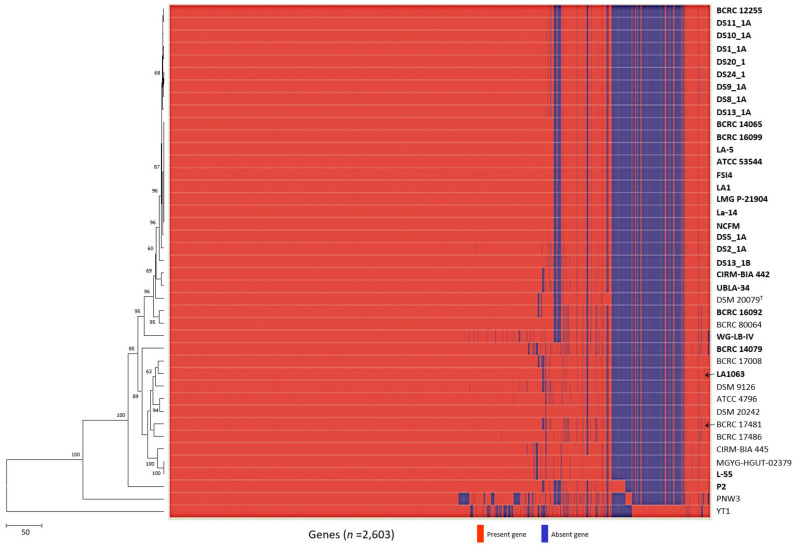
Development of the LA1063 strain-specific PCR-based identification method. Heatmap and neighbor-joining tree of the analyzed the 41 *Lactobacillus acidophilus* strains based on the presence or absence of genes. Arrows indicate strain-specific markers for *Lactobacillus acidophilus* LA1063 and BCRC 17481 strains.

**Figure 5 microorganisms-08-01445-f005:**
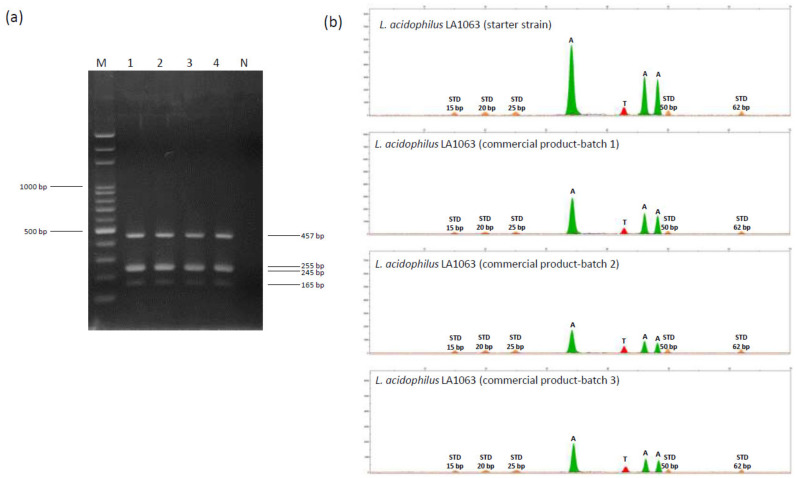
Single nucleotide polymorphism (SNP) genotyping of four polymorphisms in the *phoU*, *secY*, *tilS*, and *uvrA_*1 distinguished genes on the *Lactobacillus acidophilus* LA1063 strain. (**a**) Electropherogram of a 2% agarose gel containing multiple PCR products derived from four DNA fragments. Lane M, 100-bp ladder DNA marker; lane 1, LA1063 probiotic strain; lanes 2‒4, LA1063 isolated from separate batches of its derived probiotic product; and lane N, negative control. (**b**) Electropherograms obtained from the LA1063 strains by four-plex SNaPshot minisequencing assay. The *X*-axis represents the size of the minisequencing products (nucleotides); the *Y*-axis represents relative fluorescence units (RFUs). STD: GS120 LIZ size standard.

**Table 1 microorganisms-08-01445-t001:** Multiplex PCR primers designed to direct strain typing for *Lactobacillus acidophilus* LA1063.

Gene ^#^	Direction	Sequence (5′–3′)	Conc. (μM) ^§^	Amplicon Size (bp)
*phoU*	Forward	CATGATTATGTTCGTGCTAGA	0.03	165
	Reverse	TTCACCCGTAGTTTTGTATACC		
*uvrA**_*1	Forward	GATGACATTGCGGCTACT	0.03	245
	Reverse	GTCAAGAGTATGTCTCGCCT		
*secY*	Forward	TCGACCTTGAAGAACGCCT	0.1	255
	Reverse	CGATCTGCGCCGTAATATA		
*tilS*	Forward	TAGGACAAGCATATCGCATT	0.04	457
	Reverse	ATTGGTTCTCGATCAGCATA		

^#^*phoU*: Phosphate-specific transport system accessory protein, *uvrA_*1: UvrABC system protein A, *secY*: protein translocase, *tilS*: tRNA (Ile)-lysidine synthetase, and ^§^ final concentration in the reaction mixture.

**Table 2 microorganisms-08-01445-t002:** Single nucleotide polymorphism (SNP) primers designed to direct strain typing for *Lactobacillus acidophilus* LA1063.

Primer Name	Sequence (5′–3′) *	SNP	Conc. (μM) ^§^
SAL001390-r ^#^	GCTAAGTTAACAATGTGATCGC	A	0.2
SAL000532-f	*gactgactgactgact*GGAAAGTTATACTGGTCAATAT	A	0.3
SAL000528-f	*gactgactgac*TTTTATTCTTTTAATATACAG	T	1
SAL000608-r ^#^	*gactgactgactgactgact*TAGCTTTCACTCTAGTTCCAT	A	0.3

* Italic low-case letters indicate the nonspecific tails. ^#^ Reverse SNP primer. ^§^ Final concentration in the reaction mixture.

**Table 3 microorganisms-08-01445-t003:** List of the 92 highly discriminatory loci in 41 *Lactobacillus acidophilus* strains.

Locus	Gene	Annotation	Allele Profile																																								
			BCRC 12255	BCRC 14065	BCRC 14079	BCRC 16092	BCRC 16099	BCRC 17008	BCRC 17481	BCRC 17486	BCRC 80064	LA1063	LA-5	ATCC 4796	ATCC 53544	CIRM-BIA-442	CIRM-BIA-445	DS10_1A	DS11_1A	DS13_1A	DS13_1B	DS1_1A	DS20_1	DS24_1	DS2_1A	DS5_1A	DS8_1A	DS9_1A	DSM 20079^T^	DSM 20242	DSM 9126	MGYG-HGUT-02379	FSI4	L-55	LA1	LMG P-21904	La-14	NCFM	P2	PNW3	UBLA-34	WG-LB-IV	YT1
SAL0000004	hp	hypothetical protein	1	1	1	2	1	1	1	3	1	1	1	1	1	1	1	1	1	1	1	1	1	1	1	1	1	1	2	1	1	1	1	1	1	1	1	1	1	1	1	1	4
**SAL0000007**	**oppA_1**	**Oligopeptide-binding protein AppA precursor**	1	1	2	4	1	11	3	10	1	3	1	6	1	1	2	1	1	1	7	1	1	1	1	1	1	1	4	8	5	2	1	2	1	1	1	1	1	1	9	1	5
SAL0000024	yvgN_2	putative oxidoreductase/MSMEI_2347	1	1	3	2	1	1	1	1	1	1	1	1	1	1	1	1	1	1	1	1	1	1	1	1	4	1	2	1	1	1	1	1	1	1	1	1	1	1	1	1	5
SAL0000063	group_157	DegV domain-containing protein	1	1	2	3	6	1	5	7	1	5	1	4	1	6	2	1	1	1	1	1	1	1	1	1	1	1	3	9	4	2	1	2	8	1	1	1	1	1	1	1	1
SAL0000204	group_1709	Membrane transport protein	1	1	1	2	1	1	1	1	1	1	1	1	1	1	1	1	1	1	1	1	1	1	1	1	1	1	2	1	1	1	1	1	1	1	1	1	1	1	1	1	1
SAL0000205	pepDA_2	Dipeptidase A	1	1	2	2	1	2	2	2	1	2	1	2	1	1	2	1	1	1	1	1	1	1	1	1	1	1	3	2	2	2	1	2	4	1	1	1	1	1	1	1	5
SAL0000388	sdhB	L-serine dehydratase, beta chain	1	1	6	1	1	1	1	3	1	1	1	1	1	1	1	1	1	1	1	1	1	1	1	1	1	1	1	2	5	1	1	1	1	1	1	1	1	1	1	1	4
SAL0000391	group_1900	D-aspartate ligase	1	1	1	1	1	1	1	1	1	1	1	1	1	1	1	1	1	1	1	1	1	1	1	1	1	1	1	1	1	1	1	1	1	1	1	1	1	1	1	1	2
SAL0000409	penA	Penicillin-binding protein 2B	1	1	1	1	1	3	1	1	1	1	1	1	1	1	1	1	1	1	1	1	1	1	1	1	1	1	1	1	1	1	1	1	1	1	1	1	1	1	1	1	2
SAL0000458	group_1968	Putative phosphatase	1	1	1	1	1	1	2	1	1	2	1	1	1	1	1	1	1	1	1	1	1	1	1	1	1	1	1	1	1	1	1	1	1	1	1	1	1	1	1	1	1
SAL0000459	glyA	Serine hydroxymethyltransferase	1	1	1	1	1	1	1	1	1	1	1	1	1	1	1	1	1	1	1	1	1	1	1	1	1	1	1	1	1	1	1	1	1	1	1	1	1	1	1	1	2
SAL0000460	hp	hypothetical protein	1	1	1	2	1	1	1	2	1	1	1	1	1	1	1	1	1	1	1	1	1	1	1	1	1	1	2	1	1	1	1	1	1	1	1	1	1	1	1	1	3
SAL0000466	appA_2	Oligopeptide-binding protein AppA precursor	1	1	5	1	1	1	3	1	1	3	1	1	1	1	2	1	1	1	1	1	1	1	1	1	1	1	1	1	1	2	1	2	1	1	1	1	1	1	4	1	6
SAL0000467	group_1976	Putative tRNA (cytidine(34)-2’-O)-methyltransferase	1	1	1	1	1	1	1	1	1	1	1	1	1	1	1	1	1	1	1	1	1	1	1	1	1	1	1	1	3	1	1	1	1	1	1	1	1	1	1	1	2
SAL0000470	hslV	ATP-dependent protease subunit HslV	1	1	1	1	1	1	1	2	1	1	1	3	1	1	1	1	1	1	1	1	1	1	1	1	1	1	1	2	1	1	1	1	1	1	1	1	1	1	1	1	4
SAL0000476	ydcV	Inner membrane ABC transporter permease protein YdcV	1	1	1	1	1	1	1	1	1	1	1	3	1	1	1	1	1	1	1	1	1	1	1	1	1	1	1	1	1	1	1	1	1	1	1	1	1	1	1	1	2
SAL0000478	thrC	Threonine synthase	1	1	1	4	1	1	1	1	1	1	1	1	1	1	1	1	1	1	1	1	1	1	1	1	1	1	1	1	1	1	1	1	2	1	1	1	1	1	1	1	3
SAL0000493	citG	2-(5’’-triphosphoribosyl)-3’-dephosphocoenzyme-A synthase	1	1	1	1	1	1	1	2	1	1	1	1	1	1	1	1	1	1	1	1	1	1	1	1	1	1	1	2	1	1	1	1	1	1	1	1	1	1	1	1	3
SAL0000494	rps1	30S ribosomal protein S1	1	1	1	1	1	1	1	1	1	1	1	1	1	1	1	1	1	1	1	1	1	1	1	1	1	1	1	1	2	1	1	1	1	1	1	1	1	1	1	1	1
SAL0000501	addA	ATP-dependent helicase/nuclease subunit A	4	1	1	1	1	1	1	2	1	1	1	3	1	1	1	1	1	1	1	1	1	1	1	1	1	1	1	1	1	1	1	1	1	1	1	1	1	1	1	1	5
SAL0000512	acpS	Holo-[acyl-carrier-protein] synthase	1	1	1	1	1	1	1	1	1	1	1	1	1	1	1	1	1	1	1	1	1	1	1	1	1	1	1	1	1	1	1	1	1	1	1	1	1	1	1	2	3
**SAL0000541**	**tr**	**Transcriptional regulator**	1	1	1	1	1	1	1	1	2	1	1	1	1	1	1	2	2	2	2	1	1	2	2	2	2	2	1	1	1	1	1	1	1	3	2	1	3	2	2	1	4
SAL0000545	hp	hypothetical protein	1	1	1	1	1	1	1	3	1	1	1	1	1	1	1	1	1	1	1	1	1	1	1	1	1	1	1	1	1	1	1	1	1	1	1	1	1	1	1	1	2
SAL0000548	iscS_1	Cysteine desulfurase	1	1	1	1	1	1	1	1	1	1	1	1	1	1	1	1	1	1	1	1	1	1	1	3	1	1	1	1	1	1	1	1	1	1	1	1	1	1	1	1	2
SAL0000566	group_2084	putative hydrolase	1	1	1	1	1	1	1	2	1	1	1	4	1	1	1	1	1	1	1	1	1	3	1	1	1	1	1	1	1	1	1	1	1	1	1	5	1	1	1	1	6
SAL0000572	bca_2	C protein alpha-antigen precursor	1	1	2	1	1	1	1	7	1	1	1	5	1	1	2	1	1	3	3	1	1	1	6	1	1	1	1	1	1	2	1	2	1	1	1	1	1	1	1	1	4
SAL0000604	oppF_2	Oligopeptide transport ATP-binding protein OppF	1	1	1	1	1	1	1	1	1	1	1	1	2	1	1	1	1	1	1	1	1	1	1	1	1	1	1	1	1	1	1	1	1	1	1	1	1	1	1	1	1
SAL0000605	hp	hypothetical protein	1	1	1	1	1	1	1	1	1	1	1	1	1	1	1	1	1	1	1	1	1	1	1	1	1	1	1	1	1	1	1	1	3	1	1	1	1	1	1	1	2
SAL0000619	cdsA	Phosphatidate cytidylyltransferase	1	1	1	1	1	1	1	4	1	1	1	2	1	1	1	1	1	1	1	1	1	1	1	1	1	1	1	1	1	1	1	1	3	1	1	1	1	1	1	1	1
SAL0000652	dnaB	Replication initiation and membrane attachment protein	1	1	1	1	1	1	3	2	1	3	1	1	1	1	1	1	1	5	1	1	1	1	1	1	1	1	1	2	1	1	1	1	1	1	1	1	1	1	1	1	4
SAL0000660	hp	hypothetical protein	1	1	1	1	1	1	1	3	2	1	1	1	2	1	1	2	2	2	2	1	1	2	2	2	2	2	1	1	1	1	1	1	2	2	2	1	2	2	2	1	4
SAL0000669	pstC	Phosphate transport system permease protein PstC	1	1	1	1	1	1	2	4	1	2	1	1	1	1	1	1	1	1	1	1	1	1	1	1	1	1	1	3	1	1	1	1	1	1	1	1	1	1	1	1	1
SAL0000674	lgt	Prolipoprotein diacylglyceryl transferase	1	1	1	1	1	1	1	1	1	1	1	1	1	1	1	1	3	1	1	1	1	1	1	1	1	1	1	2	1	1	1	1	1	1	1	1	1	1	1	1	4
**SAL0000678**	**ybhL**	**Inner membrane protein YbhL**	1	1	1	5	2	1	1	3	1	1	1	10	7	2	1	1	1	1	1	1	1	1	1	1	1	1	1	8	1	1	1	1	1	1	1	4	1	1	1	9	6
SAL0000702	pepO_1	Neutral endopeptidase	1	1	2	1	1	1	1	6	1	1	1	1	1	1	1	1	1	1	1	1	1	1	1	1	1	1	1	3	1	1	1	1	1	1	1	1	1	1	5	1	4
SAL0000704	aspT	Aspartate/alanine antiporter	1	1	1	1	1	1	1	2	1	1	1	1	1	1	1	1	1	1	1	1	1	1	1	1	1	1	1	4	1	1	1	1	1	1	1	1	1	1	1	1	3
SAL0000737	corA	Magnesium transport protein CorA	1	1	2	1	1	1	1	1	1	1	1	3	1	1	2	1	1	1	1	1	1	1	1	1	1	1	1	5	1	2	1	2	1	1	1	1	1	1	1	1	4
SAL0000746	hp	hypothetical protein	1	1	1	1	1	2	1	1	1	1	1	3	1	1	1	1	1	1	1	1	1	1	1	1	1	1	1	1	1	1	1	1	1	1	1	1	1	1	1	1	1
SAL0000756	dtpT	Di-/tripeptide transporter	1	1	3	1	1	1	1	4	1	1	1	1	1	1	1	1	1	1	1	1	1	1	1	1	1	1	1	1	1	1	1	1	1	1	1	1	1	1	1	1	2
SAL0000783	lacF_1	Lactose transport system permease protein LacF	1	1	2	3	1	1	1	2	1	1	1	1	1	1	2	1	1	1	5	1	1	1	1	1	1	1	3	2	1	2	1	2	1	1	1	1	1	1	1	1	4
SAL0000796	yjbM	GTP pyrophosphokinase YjbM	1	1	1	1	1	1	1	1	1	1	1	1	1	1	1	1	1	1	1	1	1	1	1	1	1	1	1	2	1	1	1	1	1	1	1	1	1	1	1	1	3
SAL0000802	manX_1	PTS system mannose-specific EIIAB component	1	1	1	1	1	1	1	2	1	1	1	1	1	1	1	1	1	1	1	1	1	1	1	1	1	1	1	1	1	1	1	1	1	1	1	1	1	1	1	1	1
SAL0000816	glgA	Glycogen synthase	1	1	1	1	1	1	1	1	1	1	1	1	1	1	1	1	1	1	1	1	1	1	1	1	1	1	1	1	1	1	1	1	1	1	1	1	1	1	1	1	2
SAL0000831	murAB	UDP-N-acetylglucosamine 1-carboxyvinyltransferase 2	1	1	2	1	1	1	1	1	1	1	1	1	1	1	2	1	1	1	1	1	1	1	1	1	1	1	1	1	1	2	1	2	1	1	1	1	1	1	1	1	3
SAL0000882	hp	hypothetical protein	1	1	2	1	1	1	1	2	1	1	1	1	1	1	2	1	1	1	1	1	1	1	1	1	1	1	1	2	1	2	1	2	1	1	1	1	1	1	1	1	1
SAL0000889	group_2404	putative transporter YfdV	1	1	2	1	1	1	1	1	1	1	1	1	1	1	2	1	1	1	1	1	1	1	1	1	1	1	1	1	1	2	1	2	1	1	1	1	1	1	1	1	1
SAL0000894	tig	Trigger factor	1	1	2	1	1	1	1	1	1	1	1	1	1	1	3	1	1	1	1	1	1	1	1	1	1	1	1	1	1	2	1	2	4	1	1	1	1	1	1	1	1
SAL0000931	rnc	Ribonuclease 3	1	1	1	1	1	1	1	3	1	1	1	1	1	1	1	1	1	1	1	1	1	1	1	1	1	1	1	1	1	1	1	1	1	1	1	1	1	1	1	1	2
SAL0000933	citC	[Citrate [pro-3S]-lyase] ligase	1	1	1	1	1	1	1	1	2	1	1	1	1	1	1	2	2	2	2	1	1	2	2	2	2	2	1	1	1	1	1	1	1	1	2	1	1	2	2	1	3
SAL0000944	group_2459	Uracil DNA glycosylase superfamily protein	1	1	2	1	1	1	1	1	1	1	1	1	1	1	2	1	1	1	1	1	1	1	1	1	1	1	1	1	1	2	1	2	1	1	1	1	1	1	1	1	3
SAL0000947	hp	hypothetical protein	1	1	1	1	1	1	1	1	1	1	1	1	1	1	1	1	1	1	1	1	1	1	1	1	1	1	1	1	1	1	1	1	1	1	1	1	1	1	1	1	2
SAL0000970	hp	hypothetical protein	1	1	1	1	1	1	1	1	1	1	1	1	1	1	1	1	1	1	1	1	1	1	1	1	1	1	1	2	1	1	1	1	1	1	1	1	1	1	1	1	1
SAL0000985	yycI	Two-component system YycFG regulatory protein	1	1	1	1	1	1	1	1	2	1	1	1	1	1	1	2	2	2	2	1	1	2	2	2	2	2	1	1	3	1	1	1	1	1	2	1	1	2	2	1	4
SAL0000991	nudF	ADP-ribose pyrophosphatase	1	1	3	1	1	2	2	3	1	2	1	2	1	1	3	1	1	1	1	1	1	1	1	1	1	1	1	3	2	3	1	3	1	1	1	1	1	1	1	1	2
SAL0000999	group_2512	NADH dehydrogenase-like protein	1	1	1	1	1	1	1	1	1	1	2	1	1	1	1	1	1	1	1	2	2	1	1	1	1	1	1	1	1	1	2	1	1	1	1	1	1	1	1	1	3
**SAL0001061**	**frdA**	**Fumarate reductase flavoprotein subunit**	3	1	4	1	1	1	2	1	1	2	1	1	1	1	1	1	1	1	1	1	1	1	1	1	1	1	1	1	1	1	1	1	1	1	1	1	1	1	1	1	5
SAL0001093	rpsK	30S ribosomal protein S11	1	1	2	1	1	2	2	2	1	2	1	3	1	1	2	1	1	1	1	1	1	1	1	1	1	1	1	2	2	2	1	2	1	1	1	1	1	1	1	1	2
SAL0001098	odcI	Ornithine decarboxylase, inducible	1	1	2	1	1	1	1	3	1	1	1	1	1	1	1	1	1	1	1	1	1	1	1	1	1	1	1	1	1	1	1	1	1	1	1	1	1	1	1	1	4
SAL0001105	yheI_3	putative multidrug resistance ABC transporter ATP-binding/permease protein YheI	1	1	3	1	1	1	1	5	2	1	1	1	1	1	3	2	2	2	2	1	1	2	2	2	2	2	1	3	1	3	1	3	1	1	2	1	1	2	2	1	4
SAL0001111	frr	Ribosome-recycling factor	1	1	1	1	1	1	1	1	1	1	1	1	1	1	1	1	1	1	1	1	1	1	1	1	1	1	1	1	1	1	1	1	1	1	2	2	1	1	1	1	1
SAL0001114	trmD	tRNA (guanine-N(1)-)-methyltransferase	1	1	1	1	1	1	1	1	1	1	1	1	1	1	1	1	1	1	1	1	1	1	1	1	1	1	1	1	1	1	1	1	1	1	4	2	1	1	1	1	3
SAL0001120	bglA_2	6-phospho-beta-glucosidase BglA	1	1	1	1	1	1	1	1	1	1	1	2	1	1	1	1	1	1	1	1	1	1	1	1	1	1	1	1	1	1	1	1	1	1	1	1	1	1	1	1	3
SAL0001129	tfdR	HTH-type transcriptional regulator TdfR	1	1	1	2	1	1	1	1	1	1	1	4	1	1	1	1	1	1	1	1	1	1	1	1	1	1	2	1	1	1	1	1	1	1	1	1	1	1	1	1	3
SAL0001142	arlS_1	Signal transduction histidine-protein kinase ArlS	1	1	1	1	1	1	1	2	1	1	1	4	1	1	1	1	1	1	1	1	1	1	1	1	1	1	1	2	5	1	1	1	1	1	1	1	1	1	1	1	3
SAL0001144	sacX	Negative regulator of SacY activity	3	3	3	3	3	3	3	3	3	3	3	3	4	3	3	3	3	3	3	3	3	3	3	3	3	3	3	3	3	3	3	3	3	3	3	1	3	3	3	3	2
**SAL0001150**	**hp**	**hypothetical protein**	1	1	5	1	1	4	3	11	1	3	1	4	1	1	2	1	1	1	1	1	1	1	7	1	1	1	1	8	10	2	1	2	1	1	1	1	1	1	1	9	6
SAL0001155	hp	hypothetical protein	1	1	5	1	1	1	1	6	1	1	1	2	3	1	1	1	1	1	1	1	1	1	1	1	1	1	1	1	7	1	1	1	1	1	1	1	1	1	1	1	4
SAL0001159	rsmI	Ribosomal RNA small subunit methyltransferase I	1	1	1	2	1	1	1	2	1	1	1	1	1	1	1	1	1	1	1	1	1	1	1	1	1	1	2	1	1	1	1	1	1	1	1	1	1	1	1	1	3
**SAL0001160**	**rr**	**Response regulator**	1	1	7	3	1	8	1	2	1	1	1	4	1	1	2	1	1	1	1	1	1	1	1	1	1	1	3	2	5	2	1	2	1	1	1	1	1	1	6	1	1
SAL0001163	ybiR	Inner membrane protein YbiR	1	1	3	2	1	1	4	6	1	4	1	2	1	1	3	1	1	1	1	1	1	1	1	1	1	1	2	3	2	3	1	3	1	1	1	1	1	1	2	1	5
SAL0001169	ltaS1	Lipoteichoic acid synthase 1	1	1	1	1	1	5	1	3	1	1	1	4	1	1	1	1	2	1	1	1	1	1	1	1	1	1	1	1	1	1	1	1	1	1	1	1	1	1	1	6	7
SAL0001174	recX	Regulatory protein RecX	1	1	2	1	1	1	1	1	1	1	1	1	1	1	1	1	1	1	1	1	1	1	1	1	1	1	1	1	1	1	1	1	1	1	1	1	1	1	1	1	3
SAL0001177	hp	hypothetical protein	1	1	1	1	1	1	1	1	1	1	3	1	2	1	1	1	1	1	1	1	1	1	1	1	1	1	1	1	1	1	1	1	1	1	1	1	1	1	1	1	4
SAL0001181	group_590	Putative gluconeogenesis factor	1	1	1	3	1	1	1	4	1	1	1	1	1	1	2	1	1	1	1	1	1	1	1	1	1	1	1	1	1	2	1	2	1	1	1	1	1	1	1	1	5
SAL0001204	fumC	Fumarate hydratase class II	1	1	1	1	1	4	3	1	1	3	2	1	1	1	1	1	1	1	1	2	2	1	1	1	1	1	1	1	1	1	2	1	1	1	1	1	1	1	1	1	5
SAL0001210	hp	hypothetical protein	1	1	1	1	1	1	1	1	1	1	1	3	1	1	1	1	1	1	1	1	1	1	1	1	1	1	1	1	1	1	1	1	1	1	1	2	1	1	1	1	4
SAL0001216	hp	hypothetical protein	1	1	1	1	1	1	1	1	1	1	1	1	1	1	1	1	1	1	1	1	1	1	1	1	1	1	1	1	1	1	1	1	1	1	1	1	1	1	1	1	2
SAL0001231	group_644	AMP nucleosidase	1	1	1	1	1	2	1	1	1	1	1	1	1	1	1	1	1	1	1	1	1	1	1	1	1	1	1	1	1	1	1	1	1	1	1	1	1	1	1	1	1
SAL0001251	group_665	Peptidase family M23	1	1	1	1	1	1	1	1	1	1	1	1	1	1	1	1	1	1	1	2	1	1	1	1	1	1	1	1	1	1	1	1	1	1	1	1	1	1	1	1	1
SAL0001254	merA	Mercuric reductase	1	1	4	1	1	1	1	1	1	1	1	2	1	1	1	1	1	1	1	1	1	1	1	1	1	1	1	1	1	1	1	1	1	1	1	1	1	1	1	1	3
SAL0001261	group_675	Ion channel	1	1	1	1	1	1	1	1	1	1	1	1	1	1	1	1	1	1	1	1	1	1	1	1	1	1	1	1	1	1	1	1	1	1	1	3	1	1	1	1	2
SAL0001276	licC_1	Lichenan permease IIC component	1	1	1	1	1	4	1	1	1	1	1	3	1	1	1	1	1	1	1	1	1	1	1	1	1	1	1	1	2	1	1	1	1	1	1	1	1	1	1	1	1
SAL0001295	thiN	Thiamine pyrophosphokinase	1	1	1	2	3	1	1	2	1	1	1	1	1	3	1	1	1	1	1	1	1	1	1	1	1	1	1	1	1	1	1	1	1	1	1	5	1	1	1	1	4
SAL0001311	group_731	Bacterial membrane protein YfhO	1	1	1	1	1	1	2	5	1	2	1	1	1	1	4	1	1	1	1	1	1	1	1	1	1	1	1	1	1	1	1	1	1	1	1	1	1	1	1	1	3
SAL0001329	hp	hypothetical protein	2	2	1	1	1	1	1	1	1	1	2	1	1	1	1	1	1	1	1	2	2	1	1	1	1	1	1	1	1	1	2	1	1	1	1	1	1	1	1	2	3
SAL0001334	hp	hypothetical protein	1	1	1	1	1	1	1	2	1	1	1	1	1	1	1	1	1	1	1	1	1	1	1	1	1	1	1	1	1	1	1	1	1	1	1	1	1	1	1	1	1
SAL0001343	gmk_2	Guanylate kinase	1	1	1	1	1	4	1	1	1	1	1	3	1	1	1	1	1	1	1	1	1	1	1	1	1	1	1	1	1	1	1	1	1	1	2	2	1	1	1	1	1
SAL0001359	mutL	DNA mismatch repair protein MutL	1	1	1	1	1	1	2	1	1	2	1	1	1	1	1	1	1	1	1	1	1	1	1	1	1	1	1	1	4	1	1	1	1	1	1	1	1	1	1	3	5
SAL0001361	alaS	Alanine--tRNA ligase	1	1	1	1	1	1	1	1	1	1	1	1	1	1	1	1	1	1	1	1	1	1	1	1	1	1	1	1	3	1	1	1	1	1	1	1	1	1	1	2	1
SAL0001371	hp	hypothetical protein	1	1	1	1	1	1	1	1	1	1	1	1	1	1	1	1	1	1	1	1	1	1	1	1	1	1	1	1	1	1	1	1	1	1	1	2	1	1	1	1	3
**SAL0001385**	**uvrA_2**	**UvrABC system protein A**	1	1	1	2	1	1	1	1	1	1	1	1	1	1	1	1	1	1	1	1	1	1	1	1	1	1	2	1	3	1	1	1	1	1	1	4	1	1	1	1	5
**SAL0001390**	**phoU**	**Phosphate-specific transport system accessory protein**	1	1	1	1	1	1	1	2	1	5	1	1	1	1	1	1	1	1	1	1	1	1	1	1	1	1	1	4	1	1	1	1	1	1	1	1	1	1	1	1	3
Sequence type			1	2	3	4	5	6	7	8	9	10	11	12	13	5	14	9	15	16	17	18	19	20	21	22	23	9	24	25	26	27	19	27	28	29	30	31	29	9	32	33	34

The bold letters indicate the genes used for the validation of reference strains. BCRC: Bioresource Collection and Research Center.

**Table 4 microorganisms-08-01445-t004:** Strain typing of 11 reference *Lactobacillus acidophilus* strains and 1 commercial probiotic strain by multilocus sequence-typing (MLST).

Strain	Other Designation	Allele Profile								Sequence Type
		*frdA*	hp	*oppA_*1	*rr*	*tr*	*uvrA*_2	*phoU*	*ybhL*	
BCRC 10695^T^	DSM 20079^T^	1	1	1	1	1	1	1	1	1
BCRC 12255	NCIMB 701243	1	1	1	2	1	1	2	1	2
BCRC 14065	CSCC 2401	1	1	1	1	1	1	2	1	3
BCRC 14079		3	1	1	1	3	2	2	1	4
BCRC 16092	CCUG 12853	1	1	3	1	1	1	1	1	5
BCRC 16099	CIP 103600	1	1	5	1	1	2	2	1	6
BCRC 17008	ATCC 4357	1	1	1	1	2	3	2	1	7
BCRC 17009	ATCC 53544	1	1	4	1	1	1	2	1	8
BCRC 17481	JCM 1028	2	1	1	1	2	1	2	1	9
BCRC 17486	JCM 1229	4	1	1	1	2	1	2	1	10
BCRC 80064		1	2	1	1	1	1	2	1	11
LA1063 *		2	1	1	1	2	1	2	2	12

* Commercial probiotic strain.

**Table 5 microorganisms-08-01445-t005:** Strain-specific loci for *Lactobacillus acidophilus* LA1063.

Locus	Gene	Annotation
SAL0000203	*ribD*	Riboflavin biosynthesis protein ribD
SAL0000410	*purN*	Phosphoribosylglycinamide formyltransferase
**SAL0000528**	***secY***	preprotein translocase subunit secY
**SAL0000532**	***uvrA_*1**	UvrABC system protein A
**SAL0000608**	***tilS***	tRNA(Ile)-lysidine synthase
SAL0000626	group_2144	acid-resistance membrane protein
SAL0000650	*ddlA*	D-alanine--D-alanine ligase A
SAL0000698	*murF*	UDP-N-acetylmuramoyl-tripeptide--D-alanyl-D-alanine ligase
SAL0000781	*purB*	Adenylosuccinate lyase
SAL0000810	group_2329	putative peptidase
SAL0000850	*rbgA*	Ribosome biogenesis GTPase A
SAL0000942	hp	hypothetical protein
SAL0001079	*dnaI*	Primosomal protein DnaI
SAL0001095	*ybaK*	Cys-tRNA(Pro)/Cys-tRNA(Cys) deacylase YbaK
SAL0001189	*atpE*	ATP synthase subunit c
SAL0001312	*yodB*	HTH-type transcriptional regulator YodB
SAL0001323	*lldD*	L-lactate dehydrogenase [cytochrome]
SAL0001344	*mepS_2*	Murein DD-endopeptidase MepS/Murein LD-carboxypeptidase precursor
SAL0001364	group_812	Cysteine-rich secretory protein family protein
SAL0001365	*yicI_1*	Alpha-xylosidase
**SAL0001390**	***phoU***	Phosphate-specific transport system accessory protein

The bold letters indicate the genes used for developing strains-specific SNP detection methods for strain LA1063.

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
