# Peer review of "Development of a High-Resolution Single-Nucleotide Polymorphism Strain-Typing Assay Using Whole Genome-Based Analyses for the Lactobacillus acidophilus Probiotic Strain"

_microorganisms, 2020, doi:10.3390/microorganisms8091445_

Round 1
Reviewer 1 Report
The manuscript Microorganisms-913473 shows the application of a whole genome sequence–based analysis associated with a differential cgMLST approach in order to type several L. acidophilus strains and track the probiotic strain LA1063. The manuscript is well written and the experimental design shows no pitfalls or critical issue. Compared to the state of the art, the study reveals significant new advances that can be very useful in tracking strains of L. acidophilus intentionally added in supplements or food products. In this sight, the study looks like a challenge, since there are more than 29000 strains of L. acidophilus in worldwide microbial collections (http://gcm.wfcc.info/). However, it seems that the authors pay particular attention to the probiotic strain LA1063 without fully justifying their choice.
Further minor comments and suggestion are given below.
L4. One strain?
L20-26 Please, adjust font size.
L38. “out of”
L41. identify or tracking?
L89. Replace with "The 11 L. acidophilus reference strains including the probiotic LA1063"
L103. How many genome sequences were? Did they show the same level of assembly?
L135-137. These are results.
L148. Which powder samples? Were they supplements or products?
L154. The range (min-max) of contigs is too large; it is not consistent with your statement. You can report median and interquartile range or show similar ranges reported by other authors.
L167-169. The authors showed that ca. 20% of genes are unique. However, only the core genes of 38 out 41 L. acidophilus strains were processed in cgMLST analysis. Why?
L177-178. What do the authors mean by “commercial strains” and “directly cultivated strains”?
L180. "This result" instead "This finding".
L219-229. I do not find the results of authentication trials.
Figure 5. I see that you worked on identified strains, but why did you include a reference strain belonging to other species closest to L. acidophilus?
L249-474. Please, adjust font size of references.
Table S1. Please, adjust font size of caption. It is not difficult for authors to complete source column
Table S2. Please, adjust font size.
Fig.S1. What does represent the percentage values?
Author Response
Reviewer 1:
- The manuscript Microorganisms-913473 shows the application of a whole genome sequence–based analysis associated with a differential cgMLST approach in order to type several L. acidophilus strains and track the probiotic strain LA1063. The manuscript is well written and the experimental design shows no pitfalls or critical issue. Compared to the state of the art, the study reveals significant new advances that can be very useful in tracking strains of L. acidophilus intentionally added in supplements or food products. In this sight, the study looks like a challenge, since there are more than 29000 strains of L. acidophilus in worldwide microbial collections (http://gcm.wfcc.info/). However, it seems that the authors pay particular attention to the probiotic strain LA1063 without fully justifying their choice.
> Thank you for your comments. As you mentioned, a total of 29,400 L. acidophilus strain numbers are found in the “List by strain number” of WFCC Info database. However, those numbers are registered in the culture collections around the world, and almost of those numbers are overlapped of the limited number of strains. As you can see in the attached screenshot of the next page “Strain Number” in this site, a total of 141 strains are listed. However, in fact, many strain numbers are still overlapped among the culture collections, and the available strains for users are limited, too. Therefore, in this study, we have used a total of 41 L. acidophilus genomes (including 11 reference strains obtained from BCRC and a probiotic strain LA1063, and 31 whole genome sequences from NCBI, for developing the strain-specific typing methods for strain LA1063 using comparative genome analyses.
As you pointed out, we have not indicated any information about strain LA1063 in the manuscript. Therefore, we have added the brief information of it in the revised version of our manuscript (lines 92‒93).
- L4. One strain?
> Yes, we aimed mainly to develop the strain-typing method for the probiotic strain LA1063 on the basis of single-nucleotide polymorphism.
- L20-26 Please, adjust font size.
> We have corrected font size in the lines 20‒25.
- L38. “out of”
>We have corrected “of” to “out of”. (line 39)
- L41. identify or tracking?
> For “tracking” of the target strain, the accurate identification is essential for the first process. Therefore, we have chosen “identify” here (line 42).
- L89. Replace with "The 11 L. acidophilus reference strains including the probiotic LA1063"
> We have used the 11 authentic strains as references of strain LA1063 in this study. Therefore, we have kept up the original wording here (line 89).
- L103. How many genome sequences were? Did they show the same level of assembly?
>A total of 31 genomes public genome sequences of L. acidophilus strains were downloaded from the NCBI bacterial genome database. Almost all genome sequences are less than 40 contigs (median, 24; interquartile range, 17‒34), except three genome sequences (for strains DS2_1A, P2 and WG-LB-IV) showed more than 70 contigs.
8 L135-137. These are results.
> We have revised the sentence (lines 136‒138) as follows: “Strain-specific primers were designed using the genes which were chosen based on the presence or absence gene analysis and cgMLST allele profiles”.
- L148. Which powder samples? Were they supplements or products?
> We have revised as, “LA1063-derived materials for production of probiotic supplements”. (lines 145‒146)
- L154. The range (min-max) of contigs is too large; it is not consistent with your statement. You can report median and interquartile range or show similar ranges reported by other authors.
> We have revised as, (median, 24; interquartile range, 17‒34). (line 152)
- L167-169. The authors showed that ca. 20% of genes are unique. However, only the core genes of 38 out 41 L. acidophilus strains were processed in cgMLST analysis. Why?
> In our study, we found that the 1390 out of 2603 genes of 41 L. acidophilus strains were core (53.4%) on the basis of the analysis result by pan-genome analysis. Subsequent analysis by the cgMLST based on 1390 loci, we found that 38 out of 41 L. acidophilus strains were separated into two clusters, Clusters A (34 strains) and B (4 strains) (Fig. 1).
- L177-178. What do the authors mean by “commercial strains” and “directly cultivated strains”?
> “commercial strains” are the strains used in the commercial products, such as fermented milk products, probiotic supplements, etc., and “directly cultivated strains” are the strains which were isolated from the commercial products. Therefore, the both terms are equivalent. We have unified them as “commercial strains”. (line 175)
- L180."This result" instead "This finding".
> We have corrected as “This result” from “This finding”. (line 178)
- L219-229. I do not find the results of authentication trials.
> As we have described in lines 223‒226, separate batches of LA1063-derived probiotics in powder products were analyzed by using the strain-specific multiplex PCR primers and SNP primers, and we have confirmed that the nucleotide bases were found to be identical to those of the original probiotic strain (Figure 5), which demonstrated the specificity and reproducibility of this method.
- I see that you worked on identified strains, but why did you include a reference strain belonging to other species closest to L. acidophilus?
> Could you please confirm that all of the 41 strains used in this study were L. acidophilus?
- L249-474. Please, adjust font size of references.
> We have revised the font size of the references, in lines 247‒445.
- Table S1. Please, adjust font size of caption. It is not difficult for authors to complete source column
> We have revised.
- Table S2. Please, adjust font size.
> We have revised.
- Fig.S1. What does represent the percentage values?
> Average nucleotide identity (ANI) values of 95–96%, corresponds to DNA-DNA hybridization (DDH) value of 70% and has been widely using as boundary in species delineation (> 95‒96% ANI = same species).

Reviewer 2 Report
The manuscript entitled: “Development of a high-resolution single-nucleotide polymorphism strain-typing assay using whole genome–based analyses for the Lactobacillus acidophilus probiotic strain” reports information on a strain-specific single-nucleotide polymorphisms method used to identify the L. acidophilus probiotic strain LA1063. The topic fits scope and aims of the Journal. The reported data add information to the area of interest. The Introduction section should be widened and include aspects detailed in the References which are suggested to be added for better manuscript Introduction assessment and uses (see lines 53 and following).
There are reported in the following, please add:
Bartkiene E., et al., doi:10.1016/j.lwt.2018.04.017; Bartkiene E., et al., doi:10.3390/foods9040433; Bartkiene, E., et al., doi: 10.1016/j.foodcont.2016.07.010.
The experimental section seems properly assessed as well as the discussion of the results. In the Introduction a better assessment of the manuscript end points is suggested as well as in the Conclusioni it is suggested to stress perspective application of the proposed study.
A complete check of the English language is strongly suggested for better readability of the manuscript.
Author Response
Reviewer 2:
The manuscript entitled: “Development of a high-resolution single-nucleotide polymorphism strain-typing assay using whole genome–based analyses for the Lactobacillus acidophilus probiotic strain” reports information on a strain-specific single-nucleotide polymorphisms method used to identify the L. acidophilus probiotic strain LA1063. The topic fits scope and aims of the Journal. The reported data add information to the area of interest. The Introduction section should be widened and include aspects detailed in the References which are suggested to be added for better manuscript Introduction assessment and uses (see lines 53 and following).
There are reported in the following, please add: Bartkiene E., et al., doi:10.1016/j.lwt.2018.04.017; Bartkiene E., et al., doi:10.3390/foods9040433; Bartkiene, E., et al., doi: 10.1016/j.foodcont.2016.07.010.
The experimental section seems properly assessed as well as the discussion of the results. In the Introduction a better assessment of the manuscript end points is suggested as well as in the Conclusioni it is suggested to stress perspective application of the proposed study.
> Thank you very much for your valuable and constructive comments and suggestions that the Introduction section should be widened and included the aspects of probiotic properties.
As we have known well, "probiotics are live microorganisms that are intended to have health benefits when consumed or applied to the body", and their properties are strain-specific. Therefore, it is very important for us to investigate and evaluate not only the functionalities of the strains based on the scientific evidences, but also their perspective applications. Meanwhile, as we described, the end points of this study are (i) to develop a high-resolution strain-typing method for L. acidophilus probiotic strains, (ii) to establish the differential cgMLST scheme for L. acidophilus strains, and (iii) to develop the strain-specific identification method, using comparative genome analyses. This means, our article focuses only on the development of the novel taxonomic techniques.
However, your suggestion and references are extending beyond our intention, therefore, we would like to decline your comments.
A complete check of the English language is strongly suggested for better readability of the manuscript.
> We have asked the professional editing company to check and polish our manuscript. Please find its English Editing Certificate. We really hope that this revised version will now be acceptable for your standard.

Round 2
Reviewer 2 Report
The manuscript entitled: “Development of a high-resolution single-nucleotide polymorphism strain-typing assay using whole genome–based analyses for the Lactobacillus acidophilus probiotic strain” reports information on a strain-specific single-nucleotide polymorphisms method used to identify the L. acidophilus probiotic strain LA1063. The topic fits scope and aims of the Journal. The reported data add information to the area of interest. The Introduction section should be widened and include aspects detailed in the References which are suggested to be added for better manuscript Introduction assessment and uses (see lines 53 and following).
There are reported in the following, please add:
Bartkiene E., et al., doi:10.1016/j.lwt.2018.04.017; Bartkiene E., et al., doi:10.3390/foods9040433; Bartkiene, E., et al., doi: 10.1016/j.foodcont.2016.07.010.
The experimental section seems properly assessed as well as the discussion of the results. In the Introduction a better assessment of the manuscript end points is suggested as well as in the Conclusioni it is suggested to stress perspective application of the proposed study.
A complete check of the English language is strongly suggested for better readability of the manuscript.